# Intermittent Hypoxia Prevents Myocardial Mitochondrial Ca^2+^ Overload and Cell Death during Ischemia/Reperfusion: The Role of Reactive Oxygen Species

**DOI:** 10.3390/cells8060564

**Published:** 2019-06-09

**Authors:** Jui-Chih Chang, Chih-Feng Lien, Wen-Sen Lee, Huai-Ren Chang, Yu-Cheng Hsu, Yu-Po Luo, Jing-Ren Jeng, Jen-Che Hsieh, Kun-Ta Yang

**Affiliations:** 1Department of Surgery, Buddhist Tzu Chi General Hospital, Hualien 97002, Taiwan; medraytw@hotmail.com (J.-C.C.); 103327102@gms.tcu.edu.tw (Y.-P.L.); 2School of Medicine, Tzu Chi University, Hualien 97004, Taiwan; huairenchang@mail.tcu.edu.tw (H.-R.C.); jrj4511@gmail.com (J.-R.J.); jenchehsieh@gmail.com (J.-C.H.); 3Institute of Medical Sciences, Tzu Chi University, Hualien 97004, Taiwan; 98327101@gms.tcu.edu.tw; 4Graduate Institute of Medical Sciences, College of Medicine, Taipei Medical University, Taipei 11031, Taiwan; wslee@tmu.edu.tw; 5Division of Cardiology, Department of Internal Medicine, Buddhist Tzu Chi General Hospital, Hualien 97002, Taiwan; 6Master Program in Medical Physiology, School of Medicine, Tzu Chi University, Hualien 97004, Taiwan; washwolf007@gmail.com; 7Department of Physiology, School of Medicine, Tzu Chi University, Hualien 97004, Taiwan

**Keywords:** intermittent hypoxia, mitochondria, Ca^2+^, ROS, antioxidant

## Abstract

It has been documented that reactive oxygen species (ROS) contribute to oxidative stress, leading to diseases such as ischemic heart disease. Recently, increasing evidence has indicated that short-term intermittent hypoxia (IH), similar to ischemia preconditioning, could yield cardioprotection. However, the underlying mechanism for the IH-induced cardioprotective effect remains unclear. The aim of this study was to determine whether IH exposure can enhance antioxidant capacity, which contributes to cardioprotection against oxidative stress and ischemia/reperfusion (I/R) injury in cardiomyocytes. Primary rat neonatal cardiomyocytes were cultured in IH condition with an oscillating O_2_ concentration between 20% and 5% every 30 min. An MTT assay was conducted to examine the cell viability. Annexin V-FITC and SYTOX green fluorescent intensity and caspase 3 activity were detected to analyze the cell death. Fluorescent images for DCFDA, Fura-2, Rhod-2, and TMRM were acquired to analyze the ROS, cytosol Ca^2+^, mitochondrial Ca^2+^, and mitochondrial membrane potential, respectively. RT-PCR, immunocytofluorescence staining, and antioxidant activity assay were conducted to detect the expression of antioxidant enzymes. Our results show that IH induced slight increases of O_2_^−^^·^ and protected cardiomyocytes against H_2_O_2_- and I/R-induced cell death. Moreover, H_2_O_2_-induced Ca^2+^ imbalance and mitochondrial membrane depolarization were attenuated by IH, which also reduced the I/R-induced Ca^2+^ overload. Furthermore, treatment with IH increased the expression of Cu/Zn SOD and Mn SOD, the total antioxidant capacity, and the activity of catalase. Blockade of the IH-increased ROS production abolished the protective effects of IH on the Ca^2+^ homeostasis and antioxidant defense capacity. Taken together, our findings suggest that IH protected the cardiomyocytes against H_2_O_2_- and I/R-induced oxidative stress and cell death through maintaining Ca^2+^ homeostasis as well as the mitochondrial membrane potential, and upregulation of antioxidant enzymes.

## 1. Introduction

Obstructive sleep apnea (OSA), also known as intermittent hypoxia (IH), is characterized by repetitive episodic obstructions of airflow during sleep [1]. It has been indicated that IH is related to cardiovascular diseases, including hypertension, stroke, and myocardial infarction [2,3]. On the other hand, some studies have demonstrated that IH can protect against the cell death induced by ischemia or ischemia reperfusion (I/R) injury [4,5,6]. Several mechanisms have been proposed to be involved in the IH-induced protective effects, including activation of hypoxia-responsive genes, amelioration of coronary circulation, activation of protein kinase C, balance of Ca^2+^ handling activity, and inhibition of mitochondrial permeability transition pores (mPTP) opening [6,7,8,9,10]. However, the complete effects and underlying mechanisms of IH on cardiac function are still unclear.

Ca^2+^ homeostasis, which is dependent on a complex network of ion transporters, channels, and regulatory proteins, is important for maintaining cardiac functions. It has been demonstrated that the activities of L-type Ca^2+^ channel (LTCC), ryanodine receptors (RyR), sarcoplasmic/endoplasmic reticulum Ca^2+^-ATPase (SERCA) and phospholamban were regulated by Ca^2+^/calmodulin-dependent kinase II (CaMKII) and cAMP-dependent protein kinase A (PKA), and these proteins have been demonstrated to be involved in regulating intracellular Ca^2+^ homeostasis under physiological and pathophysiological conditions [11,12]. In recent years, post-translational oxidative modification of LTCC, RyR and SERCA caused by ROS also has been reported [13]. ROS has been considered an important signaling molecule in regulating cardioprotection. It has been shown that dimethylthiourea (DMTU), a potent hydroxyl radical scavenger, inhibits the ischemic preconditioning-mediated tyrosine kinase phosphorylation and cardioprotection [14]. Furthermore, the cardioprotective effects induced by diazoxide, a selective opener of the mitochondrial ATP-sensitive potassium channel, are abolished by *N*-acetyl cysteine, a ROS scavenger [15]. ROS has also been demonstrated to regulate other signaling molecules or channels, including protein tyrosine kinase, MAP kinase, phospholipase c (PLC), NFκB, IP3 receptor, ryanodine receptor, and Na^+^/Ca^2+^ exchanger (NCX) [16]. These findings suggest that ROS play a crucial role in the preconditioning-induced cardioprotection.

Intracellular Ca^2+^ plays an important role in regulating cardiomyocyte behavior under physiological and pathophysiological conditions, such as myocyte excitation‒contraction coupling, cell proliferation and differentiation, and cell death. Disruptions in Ca^2+^ handling can contribute to the pathogenesis of many diseases, such as Alzheimer’s disease, Huntington’s disease, and congestive heart failure [17]. Previous studies reported that IH can prevent the I/R-induced cell death via ameliorating Ca^2+^ homeostasis by increasing the activities of RyR, SERCA, and NCX during I/R [6,9]. I/R injury can induce burst ROS production to trigger cytosolic Ca^2+^ overload, leading to an excessive increase in mitochondrial Ca^2+^, which in turn induces mPTP to open and depolarizes the mitochondrial membrane potential. The opening of mPTP leads to a loss of ATP, mitochondrial swelling, and release of cytochrome c, resulting in apoptosis [18]. Zhu et al. reported that IH improves mitochondrial tolerance to Ca^2+^ overload and delays oxidative stress-induced mPTP opening [7]. Additionally, it has been indicated that IH can increase the expression of superoxide dismutase (SOD) and glutathione peroxidase (GPx) [19]. These findings reveal that oxidative stress might play a key role in inducing intracellular Ca^2+^ overload, and IH probably enhances the antioxidant capacity to prevent oxidative stress-induced intracellular Ca^2+^ overload. However, the relationship between IH-increased antioxidant enzyme expression and the balance of Ca^2+^ handling activity is still unclear. In this study, we aimed to determine whether IH induced cardioprotective effects through improving the intracellular Ca^2+^ balance via ROS-increased antioxidant capacity.

## 2. Materials and Methods

### 2.1. Chemicals

Fluorescent indicators were purchased from Molecular Probes (Eugene, OR, USA). Fetal bovine serum (FBS), penicillin, and trypsin were from Gibco/Life Technologies (Rockville, MD, USA). All other chemicals were purchased from Sigma (St. Louis, MO, USA).

### 2.2. Preparation of Neonatal Rat Cardiomyocytes

Neonatal rat cardiomyocytes were prepared and cultured as described previously [20]. Briefly, 1–2-day-old Sprague-Dawley rats (both sexes) were sacrificed by decapitation. The ventricles were pooled from several hearts and minced into small pieces. Cardiac tissues were digested using 0.051% pancreatin and 0.01% collagenase in Hank’s solution, and then incubated with F-12 medium containing 80% F-12 nutrient mixture, 10% horse serum, 10% FBS, and 1% penicillin (Gibco/Life Technologies) to inactivate enzymatic digestion. Cells were pre-plated on a 10-cm dish for 1 h at 37 °C in a 5% CO_2_ incubator to remove fibroblasts. Subsequently, cardiomyocytes were seeded on 24-mm collagen-coated cover slips in F-12 medium, and then added 10 μM cytosine arabinoside to inhibit fibroblast proliferation. The medium was replaced daily during the experiments. All animal studies were performed following the recommended procedures approved by the Institutional Animal Care and Use Committee of Tzu Chi University (PPL number: 104107) and conform to the guidelines from Directive 2010/63/EU of the European Parliament on the protection of animals used for scientific purposes or the NIH guidelines.

### 2.3. IH Exposures

Neonatal rat cardiomyocytes were placed in Plexiglas box chambers (length 25 cm, width 30 cm, and height 15 cm). The cardiomyocytes were exposed to room air (RA)/normoxia (20% O_2_, 5% CO_2_, and 75% N_2_) or IH (5% O_2_, 5% CO_2_, and 90% N_2_ for 30 min alternating with 30 min RA) for 1–4 days using a timer solenoid valve control [20]. Oxygen fractions in the chambers were continuously monitored by an oxygen detector. A micro dissolved oxygen electrode from Lazar Research Laboratories (DO-166MT-1, Los Angeles, CA, USA) was used to detect fluctuations in oxygen concentrations in the medium and the chamber.

### 2.4. Ischemia and Reperfusion (I/R) Injury

Simulated I/R in cultured cardiomyocytes was performed using a modified protocol described previously [21]. Briefly, cardiomyocytes were stabilized at 37 °C in Normal Tyrode (NT) buffer (140 mM NaCl, 4.5 mM KCl, 2.0 mM CaCl_2_, 1.2 mM MgCl_2_, 11 mM glucose, and 10 mM HEPES, with pH adjusted to 7.4 using NaOH) for 10 min, transferred to 100% N_2_-saturated ischemia buffer (123 mM NaCl, 8 mM KCl, 2.5 mM CaCl_2_, 0.9 mM NaH_2_PO_4_, 0.5 mM MgSO_4_, 20 mM Na-lactate, and 20 mM HEPES with pH adjusted to 6.0 using NaOH) for 6 h, and then reperfused with a culture medium for 12 h in a 5% CO_2_ incubator.

### 2.5. Analysis of Cell Death by Flow Cytometry

Apoptosis/necrosis was determined using Annexin V-FITC Apoptosis Detection Kit (BioVision, Inc., Mountain View, CA, USA) according to the manufacturer’s recommendations. Cardiomyocytes were washed with NT, dissociated by trypsin gently, harvested by centrifugation, and stained with Annexin V-FITC and SYTOX green (BioVision, Inc.) in binding buffer for 10 min at RT. Fluorescence was detected on a Gallios Flow Cytometer (Beckman Coulter, Indianapolis, IN, USA). The excitation/emission wavelengths for Annexin V-FITC and SYTOX were 488/520 nm. Based on the staining intensity, the cell population was divided into three groups: live cells (M1), apoptotic cells (M2), and necrotic cells (M3).

### 2.6. Cell Viability Assay

Cell viability was measured by the MTT assay preformed in triplicate. Briefly, cardiomyocytes were washed with NT, and incubated in culture medium containing 50 μg/mL MTT for 1 h at 37 °C in a 5% CO_2_ incubator. After removal of the MTT medium, 1 mL of DMSO was added to each dish to dissolve the precipitate for 10 min, and then detected by a Multiskan EX ELISA Reader (Thermo Scientific, Rockford, IL, USA). The absorbance at 570 nm of the control group cells was considered to be 100%.

### 2.7. Imaging of Intracellular Reactive Oxygen Species (ROS)

Intracellular levels of ROS were detected using 5-(and-6)-chloromethyl-2′,7′-dichlorodihydrofluorescein diacetate acetyl ester (CM-H_2_DCFDA; DCFDA). DCFDA fluorescence is triggered by oxidation via hydroxides (OH^−^), hydrogen peroxides (H_2_O_2_) or hydroxyl radicals (OH ^·^). Cardiomyocytes were loaded with 5 μM DCFDA for 30 min at RT, and washed twice with NT buffer. Using a real-time fluorescence imaging system including Sutter LAMBDA-LS/30 (Leica, Microsystems, Wetzlar, Germany), 1.10 set Filter wheel with Smart Shutter (Leica), and inverted fluorescent microscope (DMI3000 B; Leica) to detect fluorescence signal. Cells perfused continuously with NT buffer (37 °C) were used for fluorescent imaging. The wavelength of 480 nm was used for DCFDA excitation and 535 nm for emission. Signal increases are presented as the peak/basal fluorescence ratio (F/F_0_).

### 2.8. Flow Cytometric Analysis for Oxidative Stress Detection

Intracellular levels of ROS were detected by DCFDA. After IH or H_2_O_2_ treatment, cardiomyocytes were washed with NT solution, loaded with 5 μM DCFDA for 30 min at RT. Subsequently, cells were trypsinized, resuspended, and placed on ice. Fluorescence was measured on a Gallios Flow Cytometer (Beckman Coulter). The excitation/emission wavelengths for DCFDA was 488/540 nm (FL1).

### 2.9. Quantitative Real-Time PCR

Total RNA was isolated from neonatal rat cardiomyocytes using TRIzol reagent (Life Technologies) according to the manufacturer’s protocol. cDNA was synthesized using a Verso^TM^ cDNA Kit (Thermo). Total RNA (3 μg) was used for quantitative real-time PCR, which was performed using Maxima SYBR Green qPCR Master Mix (2×) (Thermo) with a LightCycler 480 (Roche). The primer sequences were as follows: *CuZnSOD*: forward 5′-TGG GAG AGC TTG TCA GGT G-3′, reverse 5′-CAC CAG TAG CAG GTT GCA GA-3′; *MnSOD*: forward 5′-GCC TCC CTG ACC TGC CTT AC-3′, reverse 5′-GCA TGA TCT GCG CGT TAA TG-3′; *Catalase*: forward 5′-CCC AGA AGC CTA AGA ATG CAA-3′, reverse 5′-GCT TTT CCC TTG GCA GCT ATG-3′; *GPx*: forward 5′-GTG TTC CAG TGC GCA GAT ACA-3′, reverse 5′-GGG CTT CTA TAT CGG GTT CGA-3′; and *GAPDH*: forward 5′-ATG TTC CAG TAT GAC TCC ACT CAC G-3′, reverse 5′-GAA GAC ACC AGT AGA CTC CAC GAC A-3′. All gene expression was analyzed using the comparative Ct method (2^−ΔΔCt^); ΔΔCt = ΔCt (sample) − ΔCt (reference) relative to β-actin levels.

### 2.10. Measurement of Total Antioxidant Capacity and Catalase and Glutathione Peroxidase Activity

Cardiomyocytes were scraped on ice and centrifuged at 1400× *g* at 4 °C for 10 min. The cell pellet was sonicated in assay buffer on ice, and centrifuged at 10,600× *g* at 4 °C for 15 min. Protein concentration in supernatants was quantified using the Protein Assay kit (Biorad, Hercules, CA, USA). Equal amounts of protein lysates were used for determination of total antioxidant capacity (TAC), catalase and glutathione peroxidase (GPx) activity. TAC was measured using the Antioxidant Assay Kit (Cayman Chemical, Ann Arbor, MI, USA). Catalase activity was measured using the Catalase Assay Kit (Cayman Chemical). GPx activity was measured using the Glutathione Peroxidase Assay Kit (Cayman Chemical). These assay kits were used according to the manufacturer’s instructions. 96-Well assay plates were read using an ELISA plate reader (Thermo Scientific) at 405 nm for TAC, at 540 nm for CAT, and at 340 nm for GPx.

### 2.11. Immunocytofluorescence Staining

Cardiomyocytes were fixed with methanol at 4 °C for 10 min or 10% formalin in NT at RT for 1 h, incubated in 5% non-fat milk for 60 min to block the non-specific IgG followed by primary antibody for 60 min at 37 °C, and then incubated with secondary antibody (FITC-conjugated goat anti-rabbit IgG or anti-mouse IgG) for 60 min at 37 °C. Images were obtained by confocal microscopy with resolution of 512 × 512 pixels. Green fluorescence represented FITC with excitation/emission wavelengths at 488/530 nm. MnSOD (1:200) and CuZnSOD (1:200) antibodies were purchased from Upstate Biotechnology (Lake Placid, NY, USA).

### 2.12. Imaging of Intracellular Ca^2+^ Concentrations ([Ca^2+^]_i_)

Cardiomyocytes were loaded with 5 μM Fura-2 AM for 60 min at RT, and then washed twice with NT buffer. A real-time fluorescence imaging system, including Sutter LAMBDA-LS/30 (Leica), 1.10 set Filter wheel with Smart Shutter (Leica), and inverted fluorescent microscope (DMI3000 B; Leica), was used to detect fluorescence signal. Cells perfused continuously with NT buffer (37 °C) were used for fluorescent imaging. Fura-2 was excited alternately at 340/380 nm, and emission was monitored at 520 nm. [Ca^2+^]_i_ were expressed as a ratio of F340/F380. Simulated I/R in cultured cardiomyocytes were placed on inverted fluorescent microscope. Cardiomyocytes were placed in a plexiglass box chamber containing 100% N_2_, and perfused with 100% N_2_-saturated ischemia buffer for 30 min followed by reperfusion for 30 min with 100% O_2_-saturated NT buffer.

### 2.13. Imaging of Mitochondrial Ca^2+^ Concentration ([Ca^2+^]_m_)

Changes in mitochondrial Ca^2+^ concentration were recorded using Rhod-2 AM. Cells were loaded with 5 μM Rhod-2 AM and 0.02% Pluronic F127 for 1 h at RT. The cells were also loaded with 300 nM MitoTracker Green FM (MTG) to detect the localization of mitochondria. Using time-lapse confocal laser scanning microscopy (TCS-SPII; Leica) to detect fluorescence signal. Cells perfused continuously with NT buffer (37 °C) were used for fluorescent imaging. Rhod-2 was excited at 514 nm, and emission was monitored at >530 nm. [Ca^2+^]_m_ were presented as the peak/basal fluorescence ratio (F/F_0_). MTG was excited at 488 nm, and emission was monitored at >520 nm.

### 2.14. Measurement of Mitochondrial Membrane Potential Using Flow Cytometry

Changes in mitochondrial membrane potential (ΔΨm) were recorded with tetramethylrhodamine methyl ester perchlorate (TMRM). Cells were loaded with 500 nM TMRM and Pluronic F127 for 30 min at RT. After washing with NT, cardiomyocytes were harvested, and fluorescence was measured on a FACSCalibur Flow Cytometer (Becton Dickinson Biosciences, San Jose, CA, USA). The excitation/emission wavelengths for TMRM was 488/580 nm (FL2)

### 2.15. In Situ Labeling of Activated Caspase-3

In situ labeling of activated caspase-3 was carried out using a CaspGLOW Fluorescein Active Caspase-3 Staining Kit (BioVision, Inc.). After treatment, cells were incubated with FITC–DEVD–FMK for 1 h at RT. Cardiomyocytes were fixed with 4% formaldehyde for 1 h, and then incubated with Hoechst 33342 for 15 min. Images were obtained using confocal microscopy. FITC was represented by green fluorescence with excitation/emission wavelengths at 488/>530 nm. Hoechst 33342 fluorochrome was represented by blue fluorescence with excitation/emission wavelengths at 364/400–470 nm.

### 2.16. Statistics

All results are expressed as means and standard errors of the means (mean ± SEM). Statistical differences were compared using the unpaired *t*-test, taking *p*-values < 0.05 as significant. Data obtained from three or more groups were compared by one-way ANOVA followed by LSD tests. *p*-values < 0.05 were considered significant.

## 3. Results

### 3.1. Effects of IH on Cell Death

It has been demonstrated that pathogenic or beneficial effects of IH depend on hypoxic oxygen concentration, hypoxia duration, number of cycles, and IH pattern [22]. Initially, we examined whether IH can prevent cell death. Cardiomyocytes were treated with IH for 1–4 days, followed by 30 μM H_2_O_2_ treatment for 40 min, and then the cell viability was measured using the MTT assay. As shown in Figure 1A, the viability of cells treated with four days IH (IH4) was not significantly different from the viability of those treated with RA. However, treatment with IH for 1–4 days time-dependently attenuated the H_2_O_2_-induced cell death. Furthermore, cytometric analysis for Annexin V-FITC and SYTOX green revealed that the percentage of apoptotic and necrotic cells was not significantly different between the RA-treated group and the IH4-treated group. In this study, the H_2_O_2_-treated group was used to serve as a positive control for the occurrence of apoptosis and necrosis (Figure 1B,C).

### 3.2. Effects of IH on the Endogenous Antioxidant Defense in Cardiomyocytes

Previously, we demonstrated that IH can protect cardiomyocytes against the H_2_O_2_-induced cell death [21]. Here, we further investigated whether IH can enhance the antioxidant defense capacity to against oxidative stress. To address this issue, the cells were treated with IH for four days, followed by perfusion with 30 μM H_2_O_2_ to increase the intracellular ROS level, which was detected by fluorescence probe, DCFDA, and the signals were detected by a real-time fluorescence imaging system (Figure 2A). In the RA-treated group, H_2_O_2_ treatment increased the level of intracellular ROS approximately 5-fold compared with the baseline. IH4 significantly attenuated the H_2_O_2_-induced increases of the intracellular ROS levels (Figure 2B). Next, we examined the mRNA expression levels of antioxidant enzymes by Q-PCR. As shown in Figure 2C, IH4 significantly increased the mRNA levels of Cu/Zn SOD and MnSOD. However, the mRNA levels of catalase and GPx did not change significantly. We further measured the activity of catalase, GPx and TAC using assay kit. The activities of catalase (Figure 2D), GPx (Figure 2E), and TAC (Figure 2F) were significantly higher in the IH4-treated group than the RA-treated group. These data suggest that treatment with IH for four days increased the endogenous antioxidant defense capacity.

### 3.3. Effects of IH on Oxidative Stress-Induced Intracellular Ca^2+^ Disturbance and Mitochondrial Membrane Depolarization

Previous studies have suggested that oxidative stress induces intracellular Ca^2+^ overload, which in turn causes mitochondrial membrane depolarization, hence leading cell death [23,24]. To test whether IH can prevent oxidative stress-mediated accumulation of intracellular Ca^2+^, the cardiomyocytes were treated with IH for four days followed by 30 μM H_2_O_2_ treatment to increase the intracellular ROS level, and then subjected to the intracellular Ca^2+^ concentration detection in the IH- and RA-treated cells by fluorescence probe, Fura-2. Our data show that IH4 significantly attenuated the H_2_O_2_-induced increases of the intracellular Ca^2+^ levels (Figure 3A,B). Using fluorescence probe, Rhod-2, we further demonstrated that IH4 significantly attenuated the H_2_O_2_-induced increases of the mitochondrial Ca^2+^ levels (Figure 3C,D). It has been suggested that excessive ROS can open mitochondrial permeability transition pore, hence leading mitochondrial membrane depolarization [24]. We further examined whether IH can prevent oxidative stress-mediated mitochondrial membrane depolarization. The change of mitochondrial membrane potential was evaluated by flow cytometry analysis using a fluorescence probe, TMRM, which is a cell-permeant dye that accumulates in healthy mitochondria and can be detected a strong signal. When the mitochondrial membrane depolarizes, TMRM was released from the mitochondria and weak signal was detected. We measured the average of TMRM fluorescence intensity of 10,000 cells per well. The average of TMRM in the RA group was as 100% (Figure 3E). As illustrated in Figure 3F, the TMRM signals in the IH-treated group were significantly lower than in the RA group. Under H_2_O_2_ treatment conditions, however, the IH4-treated group showed significantly higher TMRM signals than the RA group. These data demonstrate that treatment of cardiomyocytes with IH for four days significantly attenuated the H_2_O_2_-induced intracellular Ca^2+^ overload and mitochondrial membrane depolarization.

### 3.4. Involvement of ROS Generation in the IH-Increased Antioxidant Defense and Cardioprotective Effects

It has been suggested that ROS can serve as a mediator to trigger signaling pathways involved in regulating cardioprotective effects [20,21]. Flow cytometry analysis demonstrated that IH4 significantly increased the intracellular ROS generation. However, treatment of cardiomyocytes with an antioxidant, such as Apo, Phe, or MnTBAP, abolished the IH4-increased intracellular ROS generation (Figure 4A,B). We further investigated whether IH can increase antioxidant enzyme expression. Cardiomyocytes were treated with IH for four days with or without Phe (100 nM), an antioxidant, and the expression of Cu/Zn SOD and MnSOD were detected by immunocytofluorescence staining. IH4 significantly increased the expression of Cu/Zn SOD and MnSOD. However, Phe abolished the IH4-increased Cu/Zn SOD and MnSOD expression (Figure 4C,D). We also measured the total antioxidant capacity (TAC) to clarify whether IH increased endogenous antioxidant defense system via intracellular ROS generation. The levels of TAC were significantly higher in the IH4-treated group than in the RA-treated group. However, treatment of cardiomyocytes with IH for four days with Phe (100 nM) or SOD (5 U) abolished the IH4-increased TAC (Figure 4E). To investigate whether ROS was a key mediator contributing to the IH-induced cardioprotective effects, cardiomyocytes were pre-treated for four days with IH and Apo (100 μM), Phe (100 μM), or MnTBAP (50 μM) together to prevent ROS generation, followed by 30 μM H_2_O_2_ treatment for 40 min, and then subjected to measure the cell viability by MTT assay. As shown in Figure 4F, treatment with IH for four days significantly attenuated the H_2_O_2_-decreased cell viability. However, treatment with IH together with an antioxidant, such as Apo, Phe, or MnTBAP, abolished the IH-protected cell death. These data suggest that IH increased ROS generation, which upregulated antioxidant enzyme expression and increased the total antioxidant capacity, thereby preventing cell death.

### 3.5. Effects of IH on the I/R-Induced Cell Death in Cardiomyocytes

We previously demonstrated that Ca^2+^ overload plays an important role in the I/R-induced cell death [25]. In Figure 3, we demonstrated that IH4 significantly attenuated the H_2_O_2_-induced cytosolic Ca^2+^ overload. We further investigated whether IH could also prevent the I/R-induced cytosolic Ca^2+^ overload and cell death. After treatment with IH for four days, cardiomyocytes were placed in a plexiglass box chamber containing 100% N_2_, perfused with 100% N_2_-saturated ischemia buffer for 30 min, and followed by reperfusion for 30 min with 100% O_2_-saturated NT solution. Fluorescence probe, Fura-2, was used to detect the changes of Ca^2+^ during I/R. Our results show that IH4 significantly attenuated the I/R-induced increases of the cytosolic Ca^2+^ levels (Figure 5A,B). Moreover, treatment with IH for four days also significantly attenuated the I/R-decreased cell viability. However, treatment with IH for four days in the presence of an antioxidant, Apo, Phe or MnTBAP, abolished the IH-prevented cell death (Figure 5C). We also detected the level of cleaved caspase-3 to analyze apoptosis status in cardiomyocytes. As shown in Figure 5D, IH4 significantly decreased the I/R-induced apoptosis. Finally, flow cytometry analysis was conducted to measure the percentage of live cells (M1), apoptotic cells (M2) and necrotic cells (M3) in cardiomyocytes (Figure 6A). I/R increased the occurrence of apoptosis (Figure 6B) and necrosis (Figure 6C), and this effect was significantly attenuated by IH4 treatment. However, the IH-prevented apoptosis were abolished by co-treatment with Phe or MnTBAP. IH4 also significantly attenuated the I/R-induced necrosis. Noticeably, treatment with Phe or MnTBAP alone did not induce apoptosis or necrosis. These data suggest that IH prevented the I/R-induced cytosolic Ca^2+^ overload and cell death via ROS generation.

## 4. Discussion

In this study, we demonstrated that IH4 attenuated the H_2_O_2_-induced cytosolic and mitochondrial Ca^2+^ overload through ROS production, thereby preventing mitochondrial membrane depolarization. It has been reported that IH could prevent cytosolic Ca^2+^ overload through various pathways. To our knowledge, this is first demonstration that IH can protect cardiomyocytes against the H_2_O_2_- and I/R-induced oxidative stress and cell death through maintaining the Ca^2+^ homeostasis.

Gu et al. reported that IH could reduce myocardial Ca^2+^ overload and hypoxia/reoxygenation injury through upregulating PGC-1α and regulating the energy metabolism of glucose and lipid mediated by activating the HIF-1α-AMPK-PGC-1α signaling pathway [26]. HIF-1 plays a central role in cellular signaling in hypoxia adaption. It has been demonstrated that cyclic hypoxia-reoxygenation activates HIF-1 expression, which triggers expression of vascular endothelial growth factor and erythropoietin to promote angiogenesis and anti-infarct effects [10]. It has also been reported that HIF-1 could improve intracellular Ca^2+^ handling to prevent cell death. HIF-1stabilization increases protein expression of SERCA2, resulting in cytosolic Ca^2+^ reuptake into ER stores faster in thymocytes [27]. Furthermore, ischemic preconditioning activated HIF-1/Sp1 complex to increase NCX1 gene and protein expression to attenuate oxygen and glucose deprivation-induced neuronal cell death [28]. On the other hand, IH regulated the activities of Na^+^/K^+^ pump and Na^+^/Ca^2+^ exchanger to prevent Ca^2+^ overload and I/R injury through the PKC-mediated pathway has also been reported [29,30,31,32]. Additionally, IH has been shown to be able to inhibit mitochondrial Ca^2+^ overload and mitochondrial membrane depolarization, hence preventing against reperfusion injury [7,33]. Reduction of Ca^2+^ overload has been demonstrated to contribute to the IH-prevented mitochondria-mediated cell death.

During I/R injury, a burst of ROS leads to cell death, mediated by intracellular Ca^2+^ overload [34]. Excessive ROS can directly modify many Ca^2+^-handling proteins, such as RyR, SERCA, and NCX, to influence cardiomyocyte viability and heart functions [35]. We observed that treatment with IH for four days prevented the I/R- and H_2_O_2_-induced Ca^2+^ overload associated with an increase of antioxidant defense capacity through non-lethal ROS generation. IH can not only upregulate the expression of Cu/Zn SOD and MnSOD proteins but also increases catalase and GPx activity. Aguilar et al. also found that IH can increase SOD and GPx expression in cardiac tissue associated with a higher ejection and shortening fraction of the left ventricle function [19]. Increases of antioxidant capacity prevent intracellular Ca^2+^ overload through improving the intracellular Ca^2+^ handling capacity.

ROS can be both harmful and protective [36]. It has been shown that several intracellular sources including mitochondria, NADPH oxidase, xanthine oxidase, and uncoupled nitric oxide synthase can generate ROS [35]. Our data revealed that apocynin (Apo), a NADPH-oxidase inhibitor, Phenanthroline (Phe), an iron chelator that blocks the formation of hydroxyl radicals, and MnTBAP, a SOD mimetic, inhibited the IH-induced ROS generation. Our data also revealed that IH-induced ROS generation did not cause cell death (Figure 1). Furthermore, reduction of ROS generation abolished the IH-induced prevention of cell death in cardiomyocytes (Figure 5C). Estrada et al. also found that treatment of mongrel dogs with IH for 20 days induces robust cardioprotection against I/R, manifested by a 90% reduction in left ventricular infarct size. However, the cardioprotective effects induced by IH are abolished by pre-treatment with an antioxidant, *N*-acetylcysteine (NAC). They demonstrated that ROS are obligatory participants in regulating cardioprotection induced by IH [37]. It seems that non-lethal ROS may play an important role in the IH-induced cardioprotection.

In cells, large amount of ROS induces severe oxidative stress, which causes damages of DNA, lipids and proteins, thereby contributing to many different disease developments, including cardiovascular diseases [38]. However, it has also been demonstrated that ROS can serve as a redox signaling molecule in physiological conditions. Kelch-like ECH-associated protein-1 (Keap1) is a redox sensor associated with NFE2-like 2 (Nrf2). ROS oxidation of cysteines in Keap1 leads to Nrf2 release and translocation into the nucleus. Subsequently, the Nrf2-small Maf heterodimer binds to the antioxidant-responsive element (ARE), resulting in upregulation of antioxidant genes, including SOD and GPx [39,40,41]. In addition, PPARγ has been reported to be able to directly modulate the expression of several antioxidants, such as MnSOD and GPx3 [42]. Our data revealed that IH-increased non-lethal ROS generation upregulated the expression of Cu/Zn SOD and MnSOD proteins and increased catalase and GPx activity. However, the mechanisms underlying ROS-increased antioxidant enzyme expression in IH still require future investigation.

We also observed that the mRNA levels of CAT and GPx did not change (Figure 2D). However, the enzyme activity of CAT and GPx increased (Figure 2C). CAT and GPx mRNA expression was probably time-dependent [43]. It has been reported that mRNA half-lives were about several minutes to several hours. Therefore, mRNA was degraded, but the protein expression or activity still increased [44].

IH has been demonstrated to exert beneficial effects on myocardial infarction, cardiac function, arrhythmias, and coronary flow [10]. IH could cause greater left ventricle ejection fraction and fractional shortening [19]. IH also protects the myocardium from I/R injury and maintains contractility by upregulating ATF6 through the Akt-mediated signaling pathway [45]. Wang et al. reported that IH can improve post-ischemic recovery of myocardial contractile function through the Akt and the PKC-ε pathway via enhancing the production of ROS during early reperfusion [46]. Although there are many different protocols for IH treatment, the findings from the present study and other labs strongly suggest that IH can trigger multiple pathways to protect cardiomyocytes and the heart against I/R injury. In conclusion, the findings of the present study demonstrate that IH could protect cardiomyocytes against H_2_O_2_- and the I/R-induced oxidative stress and cell death through the maintenance of Ca^2+^ homeostasis, mitochondrial membrane potential, and upregulation of antioxidant enzymes via non-lethal ROS production.

## Figures and Tables

**Figure 1 cells-08-00564-f001:**
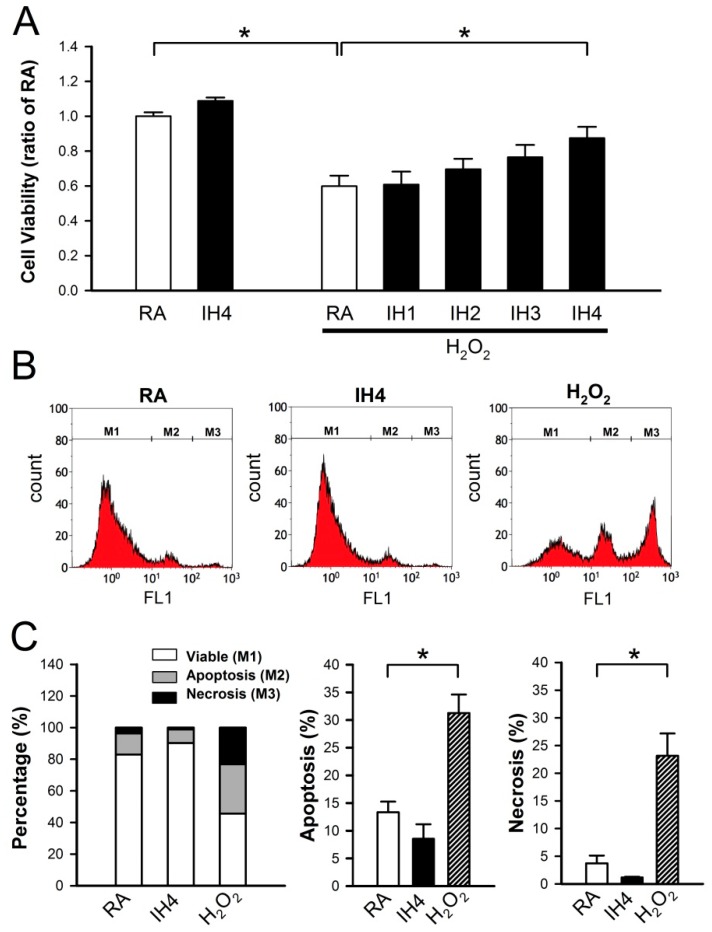
Effects of IH cell death. (**A**) Primary cultured neonatal cardiomyocytes were treated with RA or IH for 1–4 days followed by 30 μM H_2_O_2_ treatment for 40 min. Cell viability was determined using MTT assay. Treatment with IH for four days did not significantly affect the cell death. In contrast, treatment with H_2_O_2_ significantly reduced the cell viability However, treatment with IH for four days significantly reduced the H_2_O_2_-induced cell death. Data represent the means ± SEM. RA: *n* = 10, IH4: *n* = 3, RA + H_2_O_2_: *n* = 10, IH1 + H_2_O_2_: *n* = 4, IH2 + H_2_O_2_: *n* = 4, IH3 + H_2_O_2_: *n* = 4, IH4 + H_2_O_2_: *n* = 10. * *p* < 0.05. (**B**) The representative plot shows flow cytometry analysis of Annexin V-FITC and SYTOX green florescence intensity in the cells treated with RA, IH4 or H_2_O_2_. (**C**) The left panel shows the percentage of viable cells (M1), apoptotic cells (M2), and necrotic cells (M3) in the RA-, IH4-, and H_2_O_2_-treated groups. The middle panel shows that the percentage of apoptotic cells in the IH4-treated group was not significantly different from the RA-treated group. The right panel shows that the percentage of necrotic cells in the IH4-treated group was not significantly different from the RA-treated group. H_2_O_2_ treatment was serviced as a positive control for the induction of cell death. Data represent the means ± SEM. RA: *n* = 6, IH4: *n* = 3, H_2_O_2_: *n* = 6. * *p* < 0.05. RA, room air; IH4, intermittent hypoxia for four days.

**Figure 2 cells-08-00564-f002:**
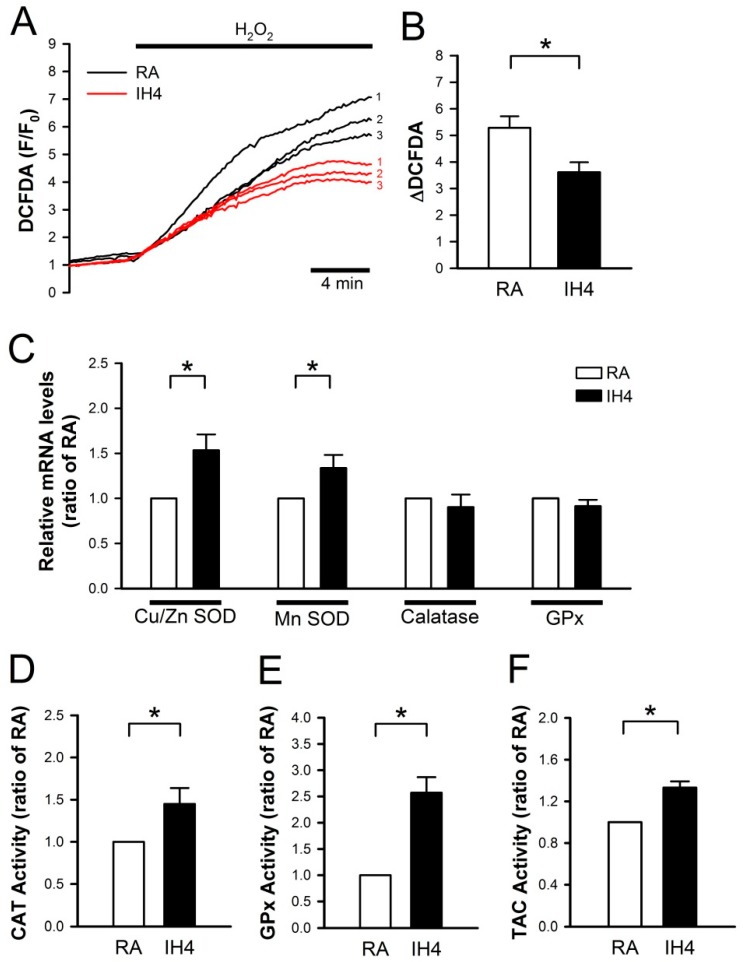
Effects of IH on endogenous antioxidant defense in cardiomyocytes. (**A**) The time course of intracellular ROS changes in cardiomyocytes perfused with H_2_O_2_ (30 μM) was recorded by a real-time fluorescence imaging system. Treatment of cardiomyocytes to IH for four days attenuated the H_2_O_2_-induced increases of intracellular ROS level. (**B**) The difference between baseline and plateau of DCFDA (F/F_0_) was defined as ΔDCFDA. The quantitative results show that the H_2_O_2_-induced intracellular ROS increase was significantly lower in cardiomyocytes treated with IH4 than with RA. Data represent the means ± SEM. *n* = 8 for each group, * *p* < 0.05. (**C**) The levels of Cu/Zn SOD and MnSOD mRNA were significantly higher in the IH4-treated group than the RA-treated group. Data represent the means ± SEM. *n* = 7 for each group. * *p* < 0.05. (**D**) The activity levels of catalase (D), GPx (**E**), and TAC (**F**) were significantly higher in the IH4-treated group than in the RA-treated group. (The activities of catalase, GPx and TAC were determined by ELISA. Data represent the means ± SEM. RA: *n* = 5, IH4: *n* = 5. * *p* < 0.05.

**Figure 3 cells-08-00564-f003:**
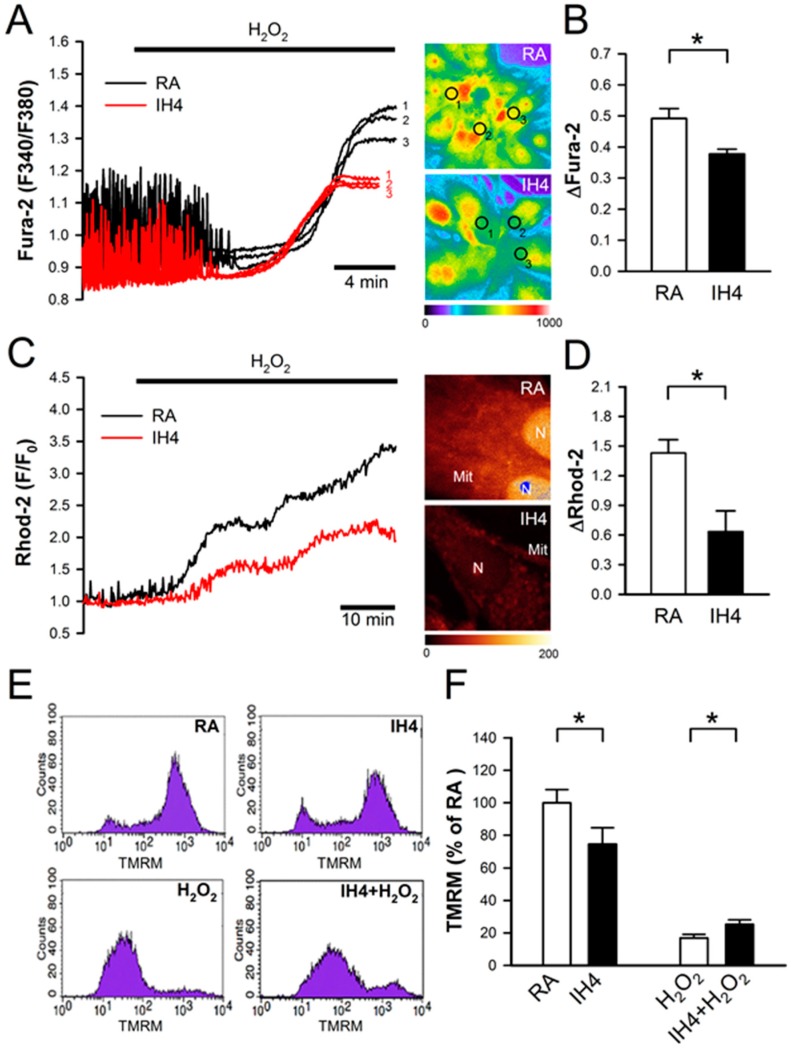
Effects of IH on H_2_O_2_-induced intracellular Ca^2+^ increase and mitochondrial membrane depolarization. (**A**) The left panel shows the time course of cytosolic Ca^2+^ changes in cardiomyocytes perfused with H_2_O_2_ (30 μM), and recorded by a real-time fluorescence imaging system. Treatment of cardiomyocytes to IH for four days attenuated H_2_O_2_-induced increases of the cytosolic Ca^2+^ level. The right panel shows representative images of cytosolic Ca^2+^ stained by Fura-2 (pseudo color). The open circle indicated the area that was examined for the intensity of Fura-2 staining. (**B**) The difference between the baseline and plateau of Fura-2 (F340/F380) was defined as ΔFura-2. The quantitative results show that the H_2_O_2_-increased cytosolic Ca^2+^ was significantly lower in the cardiomyocytes treated with IH4 than with RA. Data represent the means ± SEM. *n* = 6 for each group. * *p* < 0.05. (**C**) The left panel shows the time course of mitochondrial Ca^2+^ changes in cardiomyocytes perfused with H_2_O_2_ (30 μM), and recorded by confocal microscope. Treatment of cardiomyocytes to IH for four days attenuated the H_2_O_2_-induced increases of mitochondrial Ca^2+^ level. The right panel shows representative images of mitochondrial Ca^2+^ stained by Rhod-2 (pseudo color). (**D**) The difference between baseline and plateau of Rhod-2 (F/F_0_) was defined as ΔRhod-2. The quantitative results show that the H_2_O_2_-increased mitochondrial Ca^2+^ was significantly lower in the cardiomyocytes treated with IH4 than with RA. Data represent the means ± SEM. *n* = 3 for each group. * *p* < 0.05. (**E**) The representative plots show the mitochondrial membrane potential changes measured by flow cytometry analysis using mitochondrial membrane potential probe, TMRM. (**F**) The quantitative results are measured and normalized to RA. Mitochondrial membrane potential was significantly lower in the cardiomyocytes treated with IH4 than with RA. However, treatment of cardiomyocytes with IH for four days significantly attenuated the H_2_O_2_-induced depolarization of mitochondrial membrane potential. Data represent the means ± SEM. RA: *n* = 10, IH4: *n* = 10, H_2_O_2_: *n* = 4, IH4 + H_2_O_2_: *n* = 4. * *p* < 0.05. Mit, mitochondria; N, Nucleus.

**Figure 4 cells-08-00564-f004:**
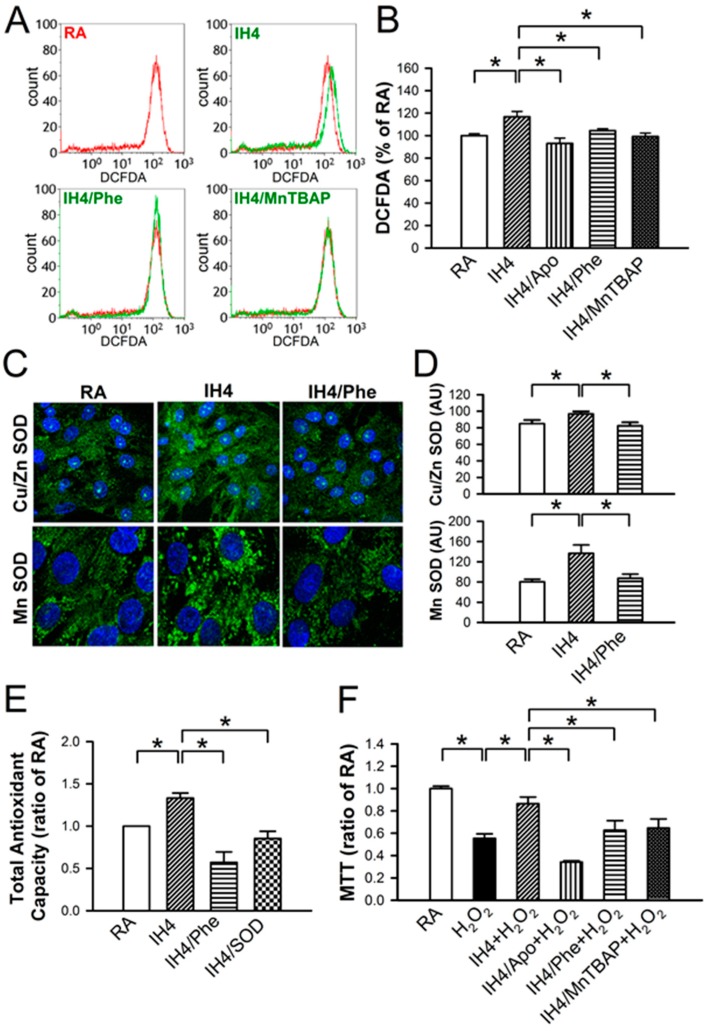
Involvement of ROS generation in the IH-increased antioxidant expression and cardioprotective effect. (**A**) The representative plots show the intracellular ROS change in various treatment measured by flow cytometry analysis using intracellular ROS probe, DCFDA. The result of cells treated with H_2_O_2_ (30 μM) in the RA was used as the positive control and shown with the red line, whereas treatment with IH4, IH4/Phe, or IH4/MnTBAP was shown with the green line. (**B**) The quantitative results show that the level of intracellular ROS was significantly higher in the cardiomyocytes treated with IH4 than with RA. However, co-treatment with an antioxidant, such as Apo, Phe, or MnTBAP, abolished the IH4-increased intracellular ROS level. Data represent the means ± SEM. RA: *n* = 10, IH4: *n* = 6, IH4/Apo: *n* = 6, IH4/Phe: *n* = 6, IH4/MnTBAP: *n* = 6. * *p* < 0.05. (**C**) The representative images show the expression of Cu/Zn SOD and MnSOD (green fluorescence) detected by immunocytofluorescence staining. The nucleus was stained by Hoechst 33342 (blue fluorescence). (**D**) The quantitative results show that the levels of Cu/Zn SOD and MnSOD were significantly higher in the cardiomyocytes treated with IH4 than with RA. However, co-treatment with Phe, an antioxidant, abolished the IH4-increased Cu/Zn SOD and MnSOD expression. Data represent the means ± SEM. (*n* = 12). * *p* < 0.05. (**E**) The levels of TAC were significantly higher in the IH4-treated group than the RA-treated group. However, co-treatment with an antioxidant, Phe or SOD, abolished the IH4-increased TAC. Total antioxidant capacity as determined by assay kit. Data represent the means ± SEM. RA: *n* = 3, IH4: = 3, IH4/Phe: *n* = 3, IH4/SOD: *n* = 3. * *p* < 0.05 (**F**) Neonatal cardiomyocytes were treated with IH for four days, and then co-treated with IH and an antioxidant, Apo, Phe or MnTBAP, followed by 30 μM H_2_O_2_ treatment for 40 min. Treatment with IH4 prevented the H_2_O_2_-induced cell death. However, co-treatment with Apo, Phe or MnTBAP abolished the IH-prevented cell death. Data represent the means ± SEM. RA: *n* = 10, RA + H_2_O_2_: *n* = 10, IH4 + H_2_O_2_: *n* = 10, IH4/Apo + H_2_O_2_: *n* = 4, IH4/Phe + H_2_O_2_: *n* = 6, IH4/MnTBAP + H_2_O_2_: *n* = 6. * *p* < 0.053.5.

**Figure 5 cells-08-00564-f005:**
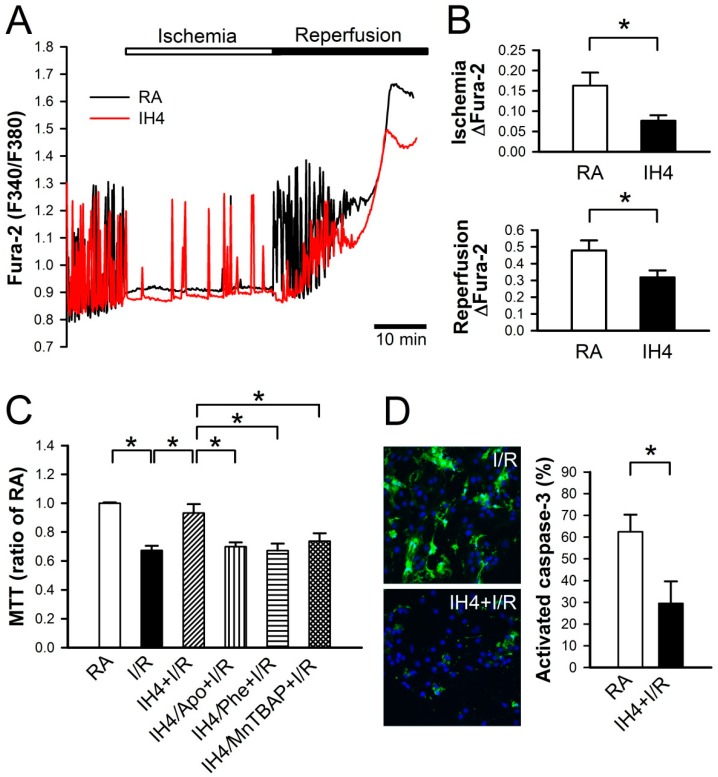
Effects of IH on the I/R-induced cytosolic Ca^2+^ increases and cell death. (**A**) The time course of cytosolic Ca^2+^ changes in cardiomyocytes perfused with ischemic and reperfusion buffer to mimic I/R injury, and recorded by a real-time fluorescence imaging system. Treatment of cardiomyocytes to IH for four days attenuated the ischemia and reperfusion-induced increases of cytosolic Ca^2+^ level. (**B**) The difference between the baseline and plateau of Fura-2 (F340/F380) was defined as ΔFura-2. The quantitative results show that the ischemia and reperfusion-increased cytosolic Ca^2+^ was significantly lower in cardiomyocytes treated with IH4 than with RA. Data represent the means ± SEM. *n* = 4 for each group. * *p* < 0.05. (**C**) Neonatal cardiomyocytes were treated with IH for four days, and then co-treated with IH and antioxidant, Apo, Phe or MnTBAP, followed by I/R treatment. Treatment with IH4 prevented the I/R-induced cell death. However, co-treatment with Apo, Phe or MnTBAP abolished the IH-prevented cell death. Data represent the means ± SEM. *n* = 5 for each group. * *p* < 0.05. (**D**) The left panel shows the representative images of activated caspase-3 expression detected by CaspGLOW fluorescein active caspase-3 staining kit after I/R treatment. The right panel gives the quantitative results, showing that the expression of active caspase-3 was significantly lower in cardiomyocytes treated with I/R + IH4 than with I/R + RA. Data represent the means ± SEM. (*n* = 14). * *p* < 0.05. I/R, ischemia/reperfusion.

**Figure 6 cells-08-00564-f006:**
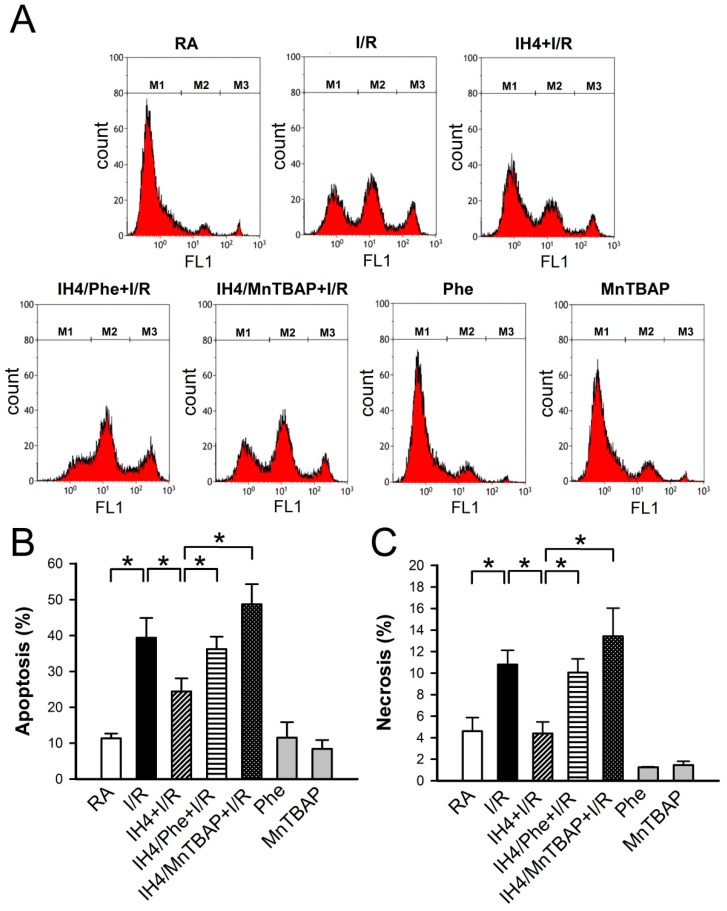
Effects of IH on the I/R-induced apoptosis and necrosis. (**A**) The representative plots show the flow cytometry analysis of Annexin V-FITC and SYTOX green florescence intensity in the cardiomyocytes treated with RA, IH4 and IH4 with an antioxidant, Phe or MnTBAP, followed by ischemia for 6 h and reperfusion for 12 h. M1, M2, and M3 represent the population of viable cells, apoptotic cells and necrotic cells, respectively. IH4 attenuated the I/R-induced apoptosis (**B**) and necrosis (**C**). IH4 with Phe or MnTBAP abolished the IH4-attenuated the I/R-induced apoptosis and necrosis.. Data represent the means ± SEM. RA: *n* = 10, RA + I/R: *n* = 6, IH4 + I/R: *n* = 5, IH4/Phe + I/R: *n* = 6, IH4/MnTBAP + I/R: *n* = 6, Phe: *n* = 4, MnTBAP: *n* = 4. * *p* < 0.05. I/R, ischemia/reperfusion; IH, intermittent hypoxia; Phe, Phenanthroline.

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
