# Peer review of "Intermittent Hypoxia Prevents Myocardial Mitochondrial Ca2+ Overload and Cell Death during Ischemia/Reperfusion: The Role of Reactive Oxygen Species"

_cells, 2019, doi:10.3390/cells8060564_

Round 1
Reviewer 1 Report
Author’s purpose is to determine how intermittent hypoxia (as a model of human obstructive nocturnal apnea) can induce cardioprotective effects on a model of newborn rat culture myocardiocytes. Their work hypothesis is that these cells after IH treatment display a better capacity in controlling cytosolic Ca++ homeostasis and that this property is acquired through a ROS-dependent mechanism.
1-Briefly, this paper is interesting as an in vitro study, the research design is adequate and methods are appropriate, although results are only partially convincing.
2-Activation of HIF1α should be analysed at different times of IH.
3-An alternative explanation for protection from Ca++homeostasis failure is that transport mechanisms (such as Na+/K+-dependent ATP-ase, Na+/Ca++-exchanger, SERCA etc.) could be potentiated by increased HIF1a-dependent transcription of many of these molecules. This alternative should be explored.
4-The potential clinical translation (see discussion) seems to me inappropriate being the in vitro experimental conditions here designed, very different and not comparable to what may occur in human obstructive nocturnal apnea and in the post-ischemic reperfusion.
In conclusion, Introduction may be improved underlining the other mechanism(s) possibly involved in controlling cytosolic/mitochondrial pCa++ homeostasis.
Research design should include the evaluation of transcription of some HIF1a-dependent gene families, especially ion transporter and exchanger.
Discussion could gain in clarity if shortened and/or divided in specific paragraphs. Moreover, statements regarding clinical translation should be avoided or smoothened.
Author Response
1. Briefly, this paper is interesting as an in vitro study, the research design is adequate and methods are appropriate, although results are only partially convincing.
Response:Thanks for your comment.
2. Activation of HIF1α should be analysed at different times of IH.
Response: Thanks for your valuable comment on the issue of HIF1aactivation at different times of IH. However, we are not able to address this issue in the present study due to time limitation (we are allowed to have 10 days for manuscript revision). We will definitely pursue this study in our future study.
We have added the relevance in the revised Discussion section as shown in below:
“HIF-1 plays a central role in cellular signaling in hypoxia adaption. It has been demonstrated that cyclic hypoxia-reoxygenation activates HIF-1 expression, which triggers expression of vascular endothelial growth factor and erythropoietin to promote angiogenesis and anti-infarct effects [10]. (page17 , lines354-357)
3. An alternative explanation for protection from Ca++homeostasis failure is that transport mechanisms (such as Na+/K+-dependent ATP-ase, Na+/Ca++-exchanger, SERCA etc.) could be potentiated by increased HIF1a-dependent transcription of many of these molecules. This alternative should be explored.
Response:As requested, we have rephrased and added the relevance in the revised Discussion section as shown in below:
“It also has been reported that HIF-1 could improve intracellular Ca2+handling to prevent cell death. HIF-1stabilization increases protein expression of SERCA2, resulting in cytosolic Ca2+reuptake into ER stores faster in thymocytes [27]. Furthermore, ischemic preconditioning activated HIF-1/Sp1 complex to increase NCX1 gene and protein expression to attenuate oxygen and glucose deprivation-induced neuronal cell death [28].” (page17 , lines357-362 )
4. The potential clinical translation (see discussion) seems to me inappropriate being the in vitro experimental conditions here designed, very different and not comparable to what may occur in human obstructive nocturnal apnea and in the post-ischemic reperfusion.
Response: We agree with your comment and rephrased the sentence as following “Although there are many different protocols for IH treatment, the findings from the present study and other labs strongly suggest the IH can trigger multiple pathways to protect cardiomyocytes and the heart against I/R injury.” (page19, lines413-415)
5. Introduction may be improved underlining the other mechanism(s) possibly involved in controlling cytosolic/mitochondrial pCa++ homeostasis.
Response:As requested, we have added the relevance in the Introduction as shown in below:
“Ca2+homeostasis, which is dependent on a complex network of ion transporters, channels and regulatory proteins, is important for maintaining cardiac functions. It has been demonstrated that the activities of L-type Ca2+channel (LTCC), ryanodine receptors (RyR), sarcoplasmic/endoplasmic reticulum Ca2+-ATPase (SERCA) and phospholamban were regulated by Ca2+/calmodulin-dependent kinase II (CaMKII) and cAMP-dependent protein kinase A (PKA), and these proteins have been demonstrated to be involved in regulating intracellular Ca2+homeostasis under physiological and pathophysiological conditions [11,12].In recent years, post-translational oxidative modification of LTCC, RyR and SERCA caused by ROS also has been reported [13].(page 3, lines 57-64)
6. Research design should include the evaluation of transcription of some HIF1a-dependent gene families, especially ion transporter and exchanger.
Response: We appreciated the reviewer’s valuable comment on the issue of evaluation of transcription of some HIF1a-dependent gene families. Again, we are not able to address this issue in the present study due to time limitation (we are allowed to have 10 days for manuscript revision). We will definitely pursue this study in our future study.
We have added the relevance in the revised Discussion section as shown in below:
Furthermore, ischemic preconditioning activated HIF-1/Sp1 complex to increase NCX1 gene and protein expression to attenuate oxygen and glucose deprivation-induced neuronal cell death [28].” (page17 , lines360-362)
7. Discussion could gain in clarity if shortened and/or divided in specific paragraphs. Moreover, statements regarding clinical translation should be avoided or smoothened.
Response:As suggested, the Discussion section has been shortened

Reviewer 2 Report
The paper by Dr. Chang et al. describes the effects of intermittent hypoxia on neonatal rat cardiomyocytes. Although the results are clear and in general sound, the are the following critical points to be addressed by the authors:
1. It is not well described how total antioxidant capacity and catalase and glutathione peroxidase activity were measured. Since as the authors claim an ELISA was used (line 155), an enzyme quantity but not the activity was most probably determined. Then the observed strong effect on enzyme levels (Fig. 2D) seems not to fit with the results of mRNA levels (Fig. 2C) where for CAT and GPx almost no change was found. That discrepancy should be at least discussed.
2. It is not well described, how changes in the mitochondrial membrane potential were detected: what means M1 and M2 in Fig. 3E, was a control measurement with total mitochondrial uncoupling performed?
3. In Fig. 4A the different colors of the histograms are not described.
Author Response
1. It is not well described how total antioxidant capacity and catalase and glutathione peroxidase activity were measured. Since as the authors claim an ELISA was used (line 155), an enzyme quantity but not the activity was most probably determined.
Response:Thanks for your comment, we have rephrased and added the relevance in the Materials and Methods as shown in below:
“Cardiomyocytes were scraped on ice and centrifuged at 1,400 ×g at 4 °C for 10 min. The cell pellet was sonicated in assay buffer on ice, and centrifuged at 10,600 ×g at 4 °C for 15 min. Protein concentration in supernatants was quantified using the Protein Assay kit (Biorad, USA). Equal amounts of protein lysates were used for determination of total antioxidant capacity (TAC), catalase and glutathione peroxidase (GPx) activity. TAC was measured using the Antioxidant Assay Kit (Cayman Chemical, USA). Catalase activity was measured using the Catalase Assay Kit (Cayman Chemical, USA). GPx activity was measured using the Glutathione Peroxidase Assay Kit (Cayman Chemical, USA). These assay kits were used according to the manufacturer's instructions. 96-Well assay plates were read using an ELISA plate reader (Thermo Scientific, USA) at 405 nm for TAC, at 540 nm for CAT, and at 340 nm for GPx.” (page 8, lines 183-192)
“ELISA kit” in page 13, line 270 have been replaced by “assay kit”.
“antioxidant capacity” in page 13, line 270 have been replaced by “activity”.
“ELISA” in page 24, line 493 have been replaced by “assay kit”.
2. The observed strong effect on enzyme levels (Fig. 2D) seems not to fit with the results of mRNA levels (Fig. 2C) where for CAT and GPx almost no change was found. That discrepancy should be at least discussed.
Response: As requested, the following information has been added in the Discussion section.
“We also observed that the mRNA levels of CAT and GPx were not changed (Fig. 2C). However, the enzyme activity of CAT and GPx were increased (Fig. 2D). CAT and GPx mRNA expression probably in a time-dependent manner [43]. It has been reported that mRNA half-lives were about several minute to several hours. Therefore, mRNA has been degraded, but the protein expression or activity still increase [44].”
(page 19, lines 402-406)
3. It is not well described, how changes in the mitochondrial membrane potential were detected: what means M1 and M2 in Fig. 3E, was a control measurement with total mitochondrial uncoupling performed?
Response:Thanks for your comment. After careful consideration, we decided to delete M1 and M2 in Fig. 3E and the sentences lines 288-293 were rephrased as following: “The change of mitochondrial membrane potential was evaluated by flow cytometry analysis using fluorescence probe, TMRM, which is a cell-permeant dye accumulated in healthy mitochondria and can be detected a strong signal. When the mitochondrial membrane depolarizes, TMRM was released from the mitochondria and weak signal was detected. We measured the average of TMRM fluorescence intensity of 10,000 cells per well. The average of TMRM in the RA group was as 100% (Fig. 3E).” (page 13, lines 288-293)
4. In Fig. 4A the different colors of the histograms are not described.
Response:As requested, We have added the color information in the revised Figure 4A legend as shown in below:
“(A) The representative plots show the intracellular ROS change in various treatment measured by flow cytometry analysis using intracellular ROS probe, DCFDA. The result of cells treated with H2O2(30 μM) in the RA was used as the positive control and shown in red line, whereas treated with IH4, IH4/Phe or IH4/MnTBAP was shown in green line.” (page 23, lines 478-481)

Round 2
Reviewer 2 Report
The authors addressed all of my comments. However, the revised version of the manuscript requires still a moderate correction of the English.